# Defending against Data-Free Model Extraction by Distributionally Robust Defensive Training

**Zhenyi Wang[1], Li Shen[2], Tongliang Liu[3], Tiehang Duan[4],**
**Yanjun Zhu[5], Donglin Zhan[6], David Doermann[7], Mingchen Gao[7]**
[1]University of Maryland, College Park, USA [2]JD Explore Academy, China
[3]The University of Sydney, Australia [4]West Virginia University, USA
[5]Northeastern University, USA [6]Columbia University, USA
[7]University at Buffalo, USA
zwang169@umd.edu; mathshenli@gmail.com; tongliang.liu@sydney.edu;
tiehang.duan@gmail.com; ya.zhu@northeastern.edu;
dz2478@columbia.edu; {doermann, mgao8}@buffalo.edu

## Abstract

Data-Free Model Extraction (DFME) aims to clone a black-box model without knowing its original training data distribution, making it much easier for attackers to steal commercial models. Defense against DFME faces several challenges: (i) effectiveness; (ii) efficiency; (iii) no prior on the attacker's query data distribution and strategy. However, existing defense methods: (1) are highly computation and memory inefficient; or (2) need strong assumptions about attack data distribution; or (3) can only *delay* the attack or *prove* a model theft after the model stealing has happened. In this work, we propose a Memory and Computation efficient defense approach, named MeCo, to *prevent* DFME from happening while maintaining the model utility simultaneously by distributionally robust defensive training on the target victim model. Specifically, we randomize the input so that it: (1) causes a mismatch of the knowledge distillation loss for attackers; (2) disturbs the zeroth-order gradient estimation; (3) changes the label prediction for the attack query data. Therefore, the attacker can only extract misleading information from the black-box model. Extensive experiments on defending against both decision-based and score-based DFME demonstrate that MeCo can significantly reduce the effectiveness of existing DFME methods and substantially improve running efficiency.

## 1 Introduction

Model extraction attack aims to replicate the functionality of a public API with only query access. Most model extraction methods focus on *data-based model extraction*, i.e., an attacker can access a small subset of the in-distribution training data [41], or a relevant surrogate dataset [36] of the target model. Beyond data-based model extraction (DBME), recent promising results with data-free model extraction (DFME) [2] show that the attacker can clone a model with performance close to that of the target black-box model even without prior knowledge of the distribution of the proprietary training data. Those DFME techniques make it much easier for attackers to steal the model without collecting and annotating relevant training data. DFME can be further categorized into *score-based* DFME [21, 51], where the target model provides softmax probability outputs to the users; and *decision-based* DFME [54, 47], where the target model only provides the top-1 label. Thus, the model owners

---

[1]Corresponding author: Zhenyi Wang, Li Shen and Mingchen Gao
[2]The problem setting name, DFME, coincides with the method name in [51]. We use DFME to denote the problem setting of model extraction without accessing the in-distribution training data, not a specific method.

37th Conference on Neural Information Processing Systems (NeurIPS 2023).

face a more critical problem - how to prevent those black-box pre-trained models from stealing in the data-free setting while simultaneously guaranteeing a high-quality inference service? The main challenges of defending against DFME are: (1) effectiveness requirement: the defender should substantially reduce the clone model accuracy; (2) efficiency requirement: the defense procedure should be memory and computation efficient; (3) lack of knowledge on the attackers: the attack query data distribution and attack strategy are unknown to the defender.

Existing work on Model Extraction (ME) defense primarily focuses on data-based ME attacks. We categorize the existing ME defense methods into pre-attack, delay-attack, and post-attack defenses. Pre-attack defenses aim to *prevent* ME attack from happening. They either perturb the output probabilities [38, 33] or integrate *multiple* models [22] or detect attack queries from benign queries [23]. Delay-attack defenses aim to *delay* the ME attack instead of preventing it [10]. Post-attack defenses neither prevent nor delay the ME; instead, they aim to *prove* a model theft [17, 18, 32]. Our work falls under the *pre-attack defense* category since our goal is to *prevent* ME from happening. However, existing pre-attack defense methods have limitations when applied to DFME: (1) output perturbation-based methods perform optimization during deployment is computationally and memory expensive; (2) ensemble-based methods are memory inefficient since they need to store multiple models; (3) detection-based methods require strong assumptions about the query data distribution; the attacker can easily make the attack queries indistinguishable from benign ones [59] and circumvent the detection [28], rendering the defense ineffective; (4) some defense methods [33, 23, 22] require the knowledge of the attack query data. However, this prior knowledge is unknown to the defender.

To address existing defense methods' limitations, we propose a new and orthogonal class of defense method, named Memory and Computation efficient defense (MeCo), through a randomized defense strategy. Specifically, MeCo adds data-dependent random perturbation to the query input. MeCo can effectively defend against DFME for several reasons. For *score-based* DFME: (i) it leads to a mismatch of the knowledge distillation loss for attackers; (ii) existing DFME methods heavily rely on zeroth-order gradient estimation. MeCo can disturb the zeroth-order gradient estimation so that the attacker can only obtain its *inaccurate* estimation. For *decision-based* DFME, MeCo can change the label prediction of attack query data. The attacker can only learn from incorrectly labeled data.

MeCo would reduce the target model utility on benign queries without additional mechanisms. Maintaining the target model utility needs to : (i) minimally sacrifice the classification accuracy on the test set, and (ii) minimize the perturbation magnitude of the output class probability. To maintain the target model utility on benign queries, we propose a principled distributionally robust optimization (DRO) framework to train the target model and perturbation generator. Our DRO framework simulates the worst-case in-distribution (ID) test data from the training data (accessible for the defender). Then, we apply the random perturbation to the simulated test data and train the target model and perturbation generator on the simulated test data to ensure the worst-case generalization on the ID test data. Compared to existing works, MeCo has numerous advantages, including (1) MeCo is substantially more computation and memory efficient without complex optimization and storing multiple models during deployment; (2) it avoids detecting attack queries from benign ones; (3) does not need the knowledge of the attack query data distribution. More importantly, even if attackers know our defense strategy and adopt an adaptive attack strategy, MeCo is still effective since stealing a *random function* further increases the difficulty of ME thanks to the randomness introduced in MeCo. Extensive experiments compared to various defense methods show that MeCo significantly reduces the accuracy of the clone model across different query budgets.

In summary, our main contributions are three-fold:

• We propose a novel principled defensive training framework that substantially improves the memory and computation efficiency during deployment to defend against DFME attacks.

• We propose a distributionally robust optimization (DRO) method to randomly perturb the inputs to defend against DFME effectively while maintaining the model utility simultaneously.

• Extensive experiments on defending against both score-based and decision-based DFME show the effectiveness of MeCo, reducing the clone model accuracy by up to $35\%$, while maintaining the target model utility. Further, MeCo can also effectively defend against data-based ME and boost its performance. More importantly, MeCo achieves substantially more computation and memory efficiency than existing methods, e.g., $17\times \sim 172\times$ speed up.

## 2 Related Work

**Data-Free Model Extraction (DFME).** Model extraction (ME) attack aims to extract and clone the functionality of the public API with only query access; representative works include [31, 9, 50, 53, 35, 36, 7, 63, 16, 40, 4, 60, 27]. Recently, ME has been extended to the data-free setting, named DFME, which cannot access original training data. ZSDB3KD [54] and DFMS-HL [47] focus on the decision-based setting, i.e., only the hard label (top-1 label) is predicted for the attacker. MAZE and DFME [21, 51] are score-based methods, i.e., soft label (softmax output) is delivered to the attacker.

**Model Extraction Defense.** We categorize the model extraction defense methods into three categories: pre-attack defenses, delay-attack defenses, and post-attack defenses, according to when the defense happens during an attack. **(i) Pre-attack defenses** aim to *prevent* the ME attack from happening. There are three classes of methods: (1) output probabilities perturbation-based methods [26, 38, 33]: Prediction poisoning (P-poison) [38] and GRAD [33] perform a complex optimization during deployment; (2) ensemble-based methods: EDM [22] integrates multiple models for diverse predictions. (3) detection-based methods: Adaptive misinformation [23], PRADA [19] and VarDetect [39] detect the attack queries from the benign queries. However, those methods have limitations when applied to DFME: (1) they [38, 33] significantly increase the computation and memory cost; (2) they [23, 19, 39] have a high risk of incorrectly classifying attack or benign queries. The attacker can easily evade the detection [28, 11], making the defense ineffective. (3) GRAD [33], Adaptive misinformation [23], and EDM [22] need to know the prior knowledge and distribution of attack query distribution, which is unknown to defenders in the data-free setting. *Our method falls under this pre-attack defense category* and addresses their limitations from a novel and orthogonal perspective. **(ii) Delay-attack defenses** aim to *delay* the ME attack instead of preventing the attack. Proof of work (PoW) [10] *delays* model stealing by significantly increasing the computation cost of query access for model extraction, e.g., solving a puzzle. Ours is fundamentally different from PoW in two aspects: (1) PoW could not prevent model stealing if the users spend more time, computation cost, money, etc. By contrast, our method is to *prevent* model stealing instead of delaying model stealing; (2) PoW needs multiple teachers to evaluate the privacy leakage of queries [42]. Our method only requires a single teacher; thus, ours is substantially more memory and computation efficient than PoW. **(iii) Post-attack defenses** aim to *prove* a model theft after a pre-trained model has been stolen, e.g., through watermark-based [1, 12] methods [17, 49], proof-of-learning [18] and dataset inference [32]. However, the post-attack defenses only perform verification of model theft but cannot prevent the model from being stolen. This requires a model owner to obtain access to the stolen model. If the stolen model is not used as API, the defender cannot verify whether the model has been stolen.

**Distributionally Robust Optimization (DRO)** DRO is a flexible and powerful optimization framework to make decisions under uncertainty [43, 55], where robustness is an important factor [62, 56]. DRO constructs a set of probability distributions, known as an ambiguity set, and then minimizes the worst-case performance within the ambiguity set, thus guaranteeing the model performance. There have been various machine learning applications of DRO, such as dealing with group-shift [45], subpopulation shift [61], and long-tailed learning [57]. To the best of our knowledge, our work is the *first* principled method with DRO for DFME defense.

## 3 Problem Setup and Preliminaries

### 3.1 Problem Setup

**Data-Free Model Extraction (Attacker).** In DFME, the attacker sends a query input $x$ to the target victim model $T$ parameterized with $\theta_T$ and receives a prediction $P(y|x) = T(x; \theta_T)$. For the score-based setting [21, 51], the target model delivers the soft label (output class probabilities) to the attacker. In the decision-based setting [54], the target model only delivers the hard label (the top-1 class) prediction to the attacker. Following [54, 21, 51], we assume the following attacker knowledge: (1) *data-free*: the attacker cannot access the original training data distribution of the target victim model. The attacker typically employs synthetic out-of-distribution (OOD) data to query the target model in an attempt to steal it. (2) *black-box*: the attacker does not know the architecture and model parameters of the target model. Given a test dataset $\mathcal{D}_{test}$ associated with the black-box model, the attacker aims to train a compact clone model $C$ with parameters $\theta_C$ that maximize the testing accuracy.

**Defense against DFME (Defender).** Following [38, 22], we assume the defender does not know: (1) whether a query is malicious or benign; (2) the attack strategies adopted by an attacker; (3) the model architecture used by an attacker. The goals of the defender are three-fold: (1) effective: minimize the test accuracy that the attacker can achieve; (2) utility preserving: maintain the accuracy on benign inputs and minimize the perturbation magnitude of the output probability; (3) efficient: the defense procedure should be memory and computation efficient.

## 3.2 Preliminaries

**Knowledge Distillation (KD).** Existing DFME methods build on top of KD [5, 14, 44, 58, 24]. Assume we have a pre-trained teacher (target) model $T$ with parameters $\boldsymbol{\theta}_T$, and a student (clone) model $C$ with parameters $\boldsymbol{\theta}_C$. Suppose the output probabilities of teacher and student for the input $\boldsymbol{x}$ are $T(\boldsymbol{x}; \boldsymbol{\theta}_T)$ and $C(\boldsymbol{x}; \boldsymbol{\theta}_C)$, respectively. The training objective of KD is as the following:

$$\mathcal{L}(\boldsymbol{x}, y) = \mathcal{L}_c(\boldsymbol{x}, y) + \alpha \mathbb{KL}(T(\boldsymbol{x}; \boldsymbol{\theta}_T), C(\boldsymbol{x}; \boldsymbol{\theta}_C)), \tag{1}$$

where $\mathcal{L}_c(\boldsymbol{x}, y)$ is the cross-entropy loss, $\mathbb{KL}$ is the KL divergence between two probability distributions, and $\alpha$ is the weighting factor. In this work, we focus on defending against score-based and decision-based DFME methods. Due to space limitations, we give a brief description of score-based DFME while placing the details of decision-based DFME in Appendix 8.

**Score-based DFME.** The representative score-based DFME works are [51, 21]. We briefly describe how they work. The attacker has a pseudo data generator $G$ parameterized by $\boldsymbol{\theta}_G$ with random vector $\boldsymbol{z}$ as input. It generates pseudo data by $\boldsymbol{x} = G(\boldsymbol{z}; \boldsymbol{\theta}_G)$, where $\boldsymbol{z} \sim N(0, \boldsymbol{I})$. Then, the attacker sends the query $\boldsymbol{x}$ to the target and clone model; they output class probabilities according to the generated pseudo data, i.e., $\boldsymbol{y}_T = T(\boldsymbol{x}; \boldsymbol{\theta}_T)$ and $\boldsymbol{y}_C = C(\boldsymbol{x}; \boldsymbol{\theta}_C)$. The attacker jointly optimizes the clone model $\mathcal{C}$ and generator $\mathcal{G}$ as below:

$$\mathcal{L}_\mathcal{C} = \mathbb{KL}(\boldsymbol{y}_T \| \boldsymbol{y}_C), \qquad \mathcal{L}_\mathcal{G} = -\mathbb{KL}(\boldsymbol{y}_T \| \boldsymbol{y}_C) \tag{2}$$

Since the target network is black-box, to backpropagate into the generator network weights, they apply zeroth-order gradient [30] on the outputs of the generator, i.e.,

$$\nabla_{\boldsymbol{\theta}_G} \mathcal{L}_G = \frac{\partial \mathcal{L}_G}{\partial \boldsymbol{\theta}_G} = \frac{\partial \mathcal{L}_G}{\partial \boldsymbol{x}} \frac{\partial \boldsymbol{x}}{\partial \boldsymbol{\theta}_G}, \qquad \text{where} \quad \frac{\partial \mathcal{L}_G}{\partial \boldsymbol{x}} = \frac{1}{m} \sum_{i=1}^{m} \frac{\mathcal{L}_G(\boldsymbol{x} + \delta \boldsymbol{\mu}_i) - \mathcal{L}_G(\boldsymbol{x})}{\delta} \boldsymbol{\mu}_i \tag{3}$$

where $\boldsymbol{\mu}_i$ is a random direction; $m$ is the number of queries; $\delta$ is a small step size. The generator $G$ and the clone model $C$ alternatively update their parameters by minimizing the loss $\mathcal{L}_\mathcal{C}$ and $\mathcal{L}_\mathcal{G}$. The attacker can obtain a clone model $C(\boldsymbol{x}; \boldsymbol{\theta}_C)$ after a certain number of training iterations.

# 4 Methodology

To defend against DFME, we propose a DRO defensive training strategy. We present the defense method in Section 4.1. We then discuss the defensive training details and deployment algorithm in Section 4.2. We illustrate how the proposed defense can defend against DFME in Section 4.3.

## 4.1 Distributionally Robust Defensive Training

The core techniques of the score-based DFME are two-fold: (1) using KD loss to match the target and clone model outputs; (2) zeroth-order gradient estimation. On the other hand, decision-based DFME mainly relies on the label prediction of the target model. We propose a random perturbation technique to make the attacker estimate misleading information. However, adding random noise to the query input strongly restricts the flexibility of random perturbation since it is difficult to maintain the model utility while effectively defending against DFME simultaneously. Furthermore, the added randomness to the input should be input-dependent since different inputs have different sensitivities to the neural network decision boundary. We thus propose a flexible random perturbation generator to learn a data-dependent random perturbation that can adaptively generate different perturbations for different inputs while maintaining the performance on benign input. First, we denote the data-dependent random perturbation generator as $h_{\boldsymbol{\omega}}(\boldsymbol{x}, \boldsymbol{\epsilon})$ parameterized by parameters $\boldsymbol{\omega}$ with query $\boldsymbol{x}$ and random noise $\boldsymbol{\epsilon}$ as input. The perturbation generator is a two-block ResNet with filter 64. It only accounts for a tiny proportion of the target model. We explain our intuition in Figure 1. Our method builds

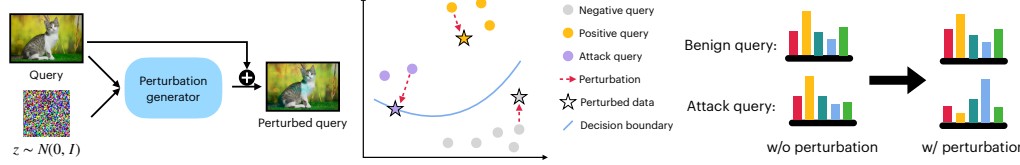

(a) Data-dependent perturbation     (b) Shift of data points     (c) Change of class probability with our defense method

Figure 1: Illustration of our defense method. (a) The perturbation generator takes the input image and random Gaussian noise as input to generate data-dependent perturbation, which is added to the original data to disturb model extraction. (b) According to [54], the attack queries are closer to the decision boundary than the benign (positive and negative) queries. All queries with added perturbation become even closer to the decision boundary, causing some attack queries to invert labels while leaving the labels of benign queries unaffected. (c) The perturbation generator generates large perturbation on attack queries so that the output class probabilities are perturbed more significantly (top-1 label may change). In contrast, the outputs on benign queries are only perturbed slightly due to the DRO defensive training.

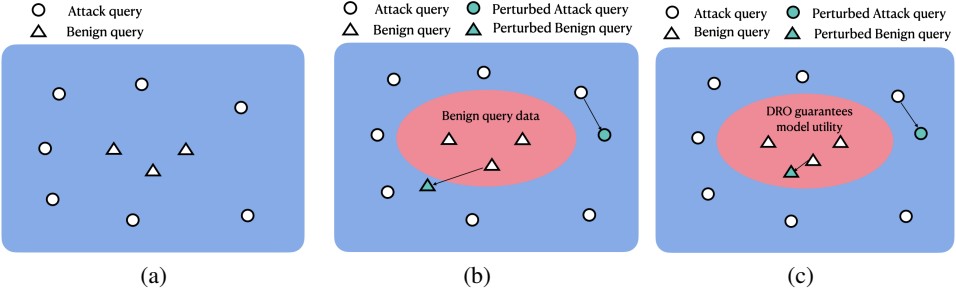

(a)            (b)            (c)

Figure 2: Illustration of our proposed defense mechanisms. (a) Using query inputs without any input perturbation results in favorable model utility but could not defend against model extraction attack. (b) The red region represents the distribution of benign query data. Applying input perturbation to every input yields excellent defense performance but compromises model utility. (c) The red region illustrates the distribution of benign query data. Distributionally robust optimization (DRO) ensures model utility by minimizing loss on the worst-case perturbed training data (simulation of test data), leading to significantly smaller perturbation magnitudes on benign inputs compared to training without DRO. Beyond the distribution of benign query data distribution (attack queries), DRO isn't employed, leading to arbitrary perturbation magnitudes on those inputs and strong defense against DFME.

on the intuition [54] that the attack query is closer to the decision boundary, while the benign query is farther away from the decision boundary in the DFME setting. We add random perturbation to the query input so that the attack query is closer to or crosses the decision boundary. Their output probabilities from the target model will be perturbed more significantly in the score-based setting. Their labels are more likely to be flipped in the decision-based setting. In contrast, benign queries are far from the decision boundary, thus not influencing benign queries much.

Simply adding the proposed data-dependent random perturbation to the query inputs would reduce the model utility. We propose a Distributionally Robust Optimization (DRO) framework to train the target model and perturbation generator to maintain the target model utility, i.e., (i) minimally sacrifice the classification accuracy on the test set, and (ii) minimize the perturbation magnitude of the output class probabilities. In addition, as depicted in Figure 2, we conducted a comparative analysis of three scenarios: one without random input perturbation, another with random input perturbation, and the third involving DRO defensive training. Our DRO defensive training exhibits a dual capability in effectively defending against DFME while maintaining the model utility.

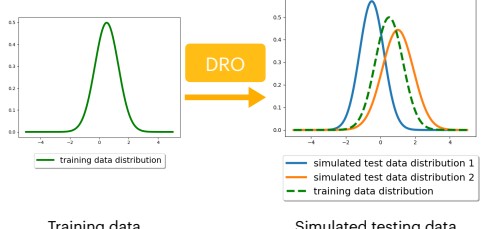

Training data        Simulated testing data

Figure 3: The proposed DRO framework perturbs original training data distribution to simulate worst-case test data distributions considering the high uncertainty of test data distribution represented by the blue and orange color on the right figure.

Since the test data distribution is unknown during deployment, we propose a flexible framework with DRO to optimize under *uncertainty* [43]. DRO constructs an ambiguity set of probability distributions and optimizes the worst-case performance within the ambiguity set, thus guaranteeing

the performance. Our framework takes that the underlying probability distribution of test data $\mu$ is *unknown* and lies in an ambiguity set of probability distributions around the training data distribution $\mu_0$. The proposed DRO framework perturbs the training data distribution to simulate the test data distribution, illustrated in Figure 3. It optimizes the worst-case performance in the ambiguity set to guarantee the model utility after adding random perturbation. We formulate the proposed DRO framework in the probability measure (distributions or densities) space :

$$\min_{\boldsymbol{\theta}_T, \boldsymbol{\omega}} \sup_{\mu \in \mathcal{P}} \mathbb{E}_{\boldsymbol{x} \sim \mu} \mathcal{L}(\boldsymbol{\theta}_T, \boldsymbol{x} + h_{\boldsymbol{\omega}}(\boldsymbol{x}, \boldsymbol{\epsilon}), y) \tag{4}$$

$$\text{s.t. } \mathcal{P} = \{\mu : \mathbb{KL}(\mu\|\mu_0) \leq \beta\} \tag{5}$$

$$\mathbb{E}_{\boldsymbol{x} \sim \mu} \|T(\boldsymbol{x} + h_{\boldsymbol{\omega}}(\boldsymbol{x}, \boldsymbol{\epsilon})) - T(\boldsymbol{x})\|_1 \leq \tau, \tag{6}$$

where the inner $\sup$ optimization in Eq. (4) is to compute and simulate the worst-case test data distribution (denoted as $\pi$) around the training data distribution (denoted as $\mu_0$). $\pi$ is defined as the probability distribution that achieves $\sup_{\mu \in \mathcal{P}} \mathbb{E}_{\boldsymbol{x} \sim \mu} \mathcal{L}(\boldsymbol{\theta}_T, \boldsymbol{x} + h_{\boldsymbol{\omega}}(\boldsymbol{x}, \boldsymbol{\epsilon}), y)$. $\mathcal{P}$ in Eq. (5) denotes the ambiguity set of probability measures for the test data distribution to characterize its uncertainty. One common choice to define $\mathcal{P}$ is through KL divergence. $\mathbb{KL}(\mu\|\mu_0)$ denotes the KL divergence between probability measure $\mu_0$ and $\mu$. $\beta$ is a constant to characterize the closeness between $\mu_0$ and $\mu$ to ensure the worst-case test data distribution $\pi$ does not deviate from the raw training data distribution $\mu_0$ much. Eq. (6) ensures that model utility (output class probabilities, i.e., $T(\boldsymbol{x})$) does not change much after adding random perturbation, i.e., $T(\boldsymbol{x} + h_{\boldsymbol{\omega}}(\boldsymbol{x}, \boldsymbol{\epsilon}))$. The probability perturbation magnitude is measured by $l_1$ norm, i.e., $\|\cdot\|_1$. It is crucial to guarantee target model output class probabilities after query perturbation close to that of without query perturbation since *benign users* need these informative probabilities to derive helpful knowledge [33]. $\tau$ is a constant.

To solve the above optimization problem, we convert Eq. (4-6) into the following unconstrained optimization problem by Lagrange duality [6] (detailed derivations are put in Appendix 11.1):

$$\min_{\boldsymbol{\theta}_T, \boldsymbol{\omega}} \sup_{\mu} [\mathbb{E}_{\boldsymbol{x} \sim \mu} \mathcal{L}(\boldsymbol{\theta}_T, \boldsymbol{x} + h_{\boldsymbol{\omega}}(\boldsymbol{x}, \boldsymbol{\epsilon}), y) - \mathbb{KL}(\mu\|\mu_0)] + \gamma \mathbb{E}_{\boldsymbol{x} \sim \mu} \|T(\boldsymbol{x} + h_{\boldsymbol{\omega}}(\boldsymbol{x}, \boldsymbol{\epsilon})) - T(\boldsymbol{x})\|_1 \tag{7}$$

The constant $\gamma$ controls the regularization magnitude for model utility. The KL-divergence $\mathbb{KL}(\mu\|\mu_0)$ is handled by Wasserstein gradient flow (WGF) (presented in the following sections), therefore for simplicity, the regularization weight for it is set to 1.0 throughout the paper. We name Eq. (7) as **Defense DRO**. The optimization in Eq. (7) is still challenging to solve as the inner $\sup$ optimization is over probability measure space, which is an infinite-dimensional *function space*.

**Solution to the Defense DRO.** To make the solution of the Defense DRO (Eq. (7)) tractable, we reformulate it from a new continuous dynamics perspective. To achieve this goal, we first define the energy functional $F(\mu)$ as follows:

$$F(\mu) = V(\mu) + \mathbb{KL}(\mu\|\pi) \tag{8}$$

where $V(\mu) = -\mathbb{E}_{\boldsymbol{x} \sim \mu} \mathcal{L}(\boldsymbol{\theta}_T, \boldsymbol{x} + h_{\boldsymbol{\omega}}(\boldsymbol{x}, \boldsymbol{\epsilon}), y)$. By defining such energy functional $F(\mu)$, the Eq. (7) can be equivalently formulated by the following gradient flow system Eq. (9, 10):

$$\begin{cases} \partial_t \mu_t &= \nabla_{W_2} F(\mu_t) := div\left(\mu_t \nabla \frac{\delta F}{\delta \mu}(\mu_t)\right); \tag{9} \\ \dfrac{d\boldsymbol{\theta}_T}{dt} &= -\nabla_{\boldsymbol{\theta}_T}[\mathbb{E}_{\boldsymbol{x} \sim \mu_t} \mathcal{L}(\boldsymbol{\theta}_T, \boldsymbol{x} + h_{\boldsymbol{\omega}}(\boldsymbol{x}, \boldsymbol{\epsilon}), y) + \gamma \mathbb{E}_{\boldsymbol{x} \sim \mu_t} \|T(\boldsymbol{x} + h_{\boldsymbol{\omega}}(\boldsymbol{x}, \boldsymbol{\epsilon})) - T(\boldsymbol{x})\|_1], \tag{10} \end{cases}$$

where Eq. (9) solves the inner $\sup$ problem in Eq. (7) with WGF in Wasserstein space (Details provided in Appendix 11.2) and Eq. (10) solves the outer minimization problem in Eq. (7) for parameter update with gradient flow in Euclidean space. Below, we propose a method for efficiently solving the Eq. (9). We view each training data as one particle and arrive at the following test data distribution simulation solution (solution to Eq. (9)) (Details are provided in Appendix 11.2):

$$\boldsymbol{x}_{t+1}^i - \boldsymbol{x}_t^i = -\frac{\alpha}{N} \sum_{j=1}^{j=N} [k(\boldsymbol{x}_t^i, \boldsymbol{x}_t^j) \nabla_{\boldsymbol{x}_t^j} U(\boldsymbol{x}_t^j, \boldsymbol{\theta}_T) + \nabla_{\boldsymbol{x}_t^j} k(\boldsymbol{x}_t^i, \boldsymbol{x}_t^j)] \tag{11}$$

where $\boldsymbol{x}_t^i$ denotes the $i^{th}$ training data $\boldsymbol{x}^i$ perturbed at time $t$; $\alpha$ is the data transformation rate. $U(\boldsymbol{x}, \boldsymbol{\theta}_T) = -\mathcal{L}(\boldsymbol{\theta}_T, \boldsymbol{x}, y)$. $k(\boldsymbol{x}_i, \boldsymbol{x}_j)$ is the Gaussian kernel, i.e., $k(\boldsymbol{x}_i, \boldsymbol{x}_j) = \exp(-\frac{(\boldsymbol{x}_i - \boldsymbol{x}_j)^2}{2\sigma^2})$. The first term in the RHS of Eq. (11) moves the training data towards the worst-case simulated test data distribution by raising the energy functional. The data distribution change is controlled by the sum of gradients from the mini-batch of the training data, which are weighted by the kernel function $k(\boldsymbol{x}_t^i, \boldsymbol{x}_t^j)$, smoothing the gradients of training data. The second term (kernel gradient) acts as a repelling force that keeps the transformed data from concentrating on a single data point, thus diversifying the simulated test data.

## 4.2 End-to-end Defensive Training for DFME

We summarize the defensive training algorithm in Algorithm 1. Line 3-4 simulates the worst-case test data distribution. Line 5-8 adds data-dependent random perturbation to the simulated test data and trains the target model and perturbation generator on the simulated test data.

**MeCo Deployment.** During testing, we perform similar perturbation for each query. Given any query input $\boldsymbol{x}$, $\boldsymbol{y}_p = T_{\boldsymbol{\theta}_T}(\boldsymbol{x} + h_{\boldsymbol{\omega}}(\boldsymbol{x}, \boldsymbol{\epsilon}))$; $\boldsymbol{\epsilon} \sim N(0, \boldsymbol{I})$. Then, return the results $\boldsymbol{y}_p$ to the user.

---

**Algorithm 1** MeCo **Training**.

---

1: **REQUIRE:** Target model $T$ with parameters $\boldsymbol{\theta}_T$; data-dependent perturbation generator $h_{\boldsymbol{\omega}}$ with parameters $\boldsymbol{\omega}$; target model learning rate $\eta$; $Q$ is the number of test data simulation steps; $\mathcal{D}_{tr}$ is the training dataset.
2: **for** $k = 1$ to $M$ **do** {$M$ is the number of training iterations}
3:     randomly sample a new mini-batch data $(\boldsymbol{x}, y)$ from $\mathcal{D}_{tr}$
4:     perturb mini-batch data to be $(\boldsymbol{x}_Q, y)$ $Q$ steps by Eq. (11).
5:     add random perturbation to $(\boldsymbol{x}_Q, y)$ to be $(\boldsymbol{x}_Q + h_{\boldsymbol{\omega}}(\boldsymbol{x}_Q, \boldsymbol{\epsilon}), y)$
6:     calculate the loss $g(\boldsymbol{\theta}_T, \boldsymbol{\omega}) = \mathcal{L}(\boldsymbol{\theta}_T^k, \boldsymbol{x}_Q + h_{\boldsymbol{\omega}}(\boldsymbol{x}_Q, \boldsymbol{\epsilon}), y) + \gamma ||T(\boldsymbol{x} + h_{\boldsymbol{\omega}}(\boldsymbol{x}, \boldsymbol{\epsilon})) - T(\boldsymbol{x})||_1$
7:     train the target model by $\boldsymbol{\theta}_T^{k+1} = \boldsymbol{\theta}_T^k - \eta \nabla_{\boldsymbol{\theta}_T}[g(\boldsymbol{\theta}_T, \boldsymbol{\omega})]$
8:     train the perturbation generator by $\boldsymbol{\omega}^{k+1} = \boldsymbol{\omega}^k - \eta \nabla_{\boldsymbol{\omega}}[g(\boldsymbol{\theta}_T, \boldsymbol{\omega})]$
9: **end for**

---

## 4.3 Why Can MeCo Defend against DFME

Below, we give explanations on why Algorithm 1 could intrinsically defend against both score-based and decision-based DFME from the perspective: (1) mismatch the KD loss for the attacker; (2) disturb the zeroth-order gradient estimation; (3) change the label prediction on attack query.

**Mismatch the KD learning objective.** The KL-divergence loss Eq. (1) is critical to match the probability output between the target and clone model. By adding data-dependent random perturbation, we can encourage the mismatch of the KD loss between the target model and clone model, i.e.,

$$\mathbb{KL}(T(\boldsymbol{x} + h_{\boldsymbol{\omega}}(\boldsymbol{x}, \boldsymbol{\epsilon}); \boldsymbol{\theta}_T), C(\boldsymbol{x}; \boldsymbol{\theta}_C)) \tag{12}$$

The mismatch would mislead the attacker to learn in the wrong direction since they think that $T(\boldsymbol{x} + h_{\boldsymbol{\omega}}(\boldsymbol{x}, \boldsymbol{\epsilon}); \boldsymbol{\theta}_T)$ is the desired output for $\boldsymbol{x}$ but should be $T(\boldsymbol{x}, \boldsymbol{\theta}_T)$, as illustrated in Figure 1.

**Disturb the zeroth-order gradient estimation.** By adding data-dependent random perturbation, the zeroth-order gradient estimation of score-based DFME ($\frac{\partial \mathcal{L}_G}{\partial \boldsymbol{x}}$ in Eq. (3)) becomes:

$$\frac{\partial \mathcal{L}_G}{\partial \boldsymbol{x}} = \frac{1}{m} \sum_{i=1}^{m} \frac{\mathcal{L}_G(\boldsymbol{x} + \delta \boldsymbol{\mu}_i + \boldsymbol{v}_i^1) - \mathcal{L}_G(\boldsymbol{x} + \boldsymbol{v}_i^2)}{\delta} \boldsymbol{\mu}_i, \tag{13}$$

where $\boldsymbol{v}_i^1$ and $\boldsymbol{v}_i^2$ are two data-dependent perturbations correspond to the two model inputs $\boldsymbol{x} + \delta \boldsymbol{\mu}_i$ and $\boldsymbol{x}$, respectively. Compared with Eq. (3), Eq. (13) would make inaccurate gradient estimation by changing the gradient estimation direction due to the inconsistent random perturbation added to different inputs. Consequently, this would lead the gradient of the pseudo data generator in the wrong direction, i.e., $\nabla_{\boldsymbol{\theta}_G} \mathcal{L}_G$ in Eq. (3) would be inaccurate. Then, the generator would not generate informative samples. The uninformative pseudo samples would cause the clone model Eq. (2) to learn incorrect information.

Therefore, by jointly (i) mismatching the KD loss for the attacker; (ii) disturbing the zeroth-order gradient estimation, MeCo can effectively defend against DFME. In summary, MeCo has many advantages compared to existing methods since it *does not need to*: (1) solve a complex optimization during testing; (2) store multiple models for ensemble; (3) detect the attack query from the benign query; (4) know the attack query data distribution. Due to the space limitations, we additionally show why MeCo can defend against decision-based DFME methods by changing the label prediction on attack query in Appendix 8.

**Theoretical Analysis** In this section, we delve into the theoretical analysis of our proposed method. We assess the effectiveness of our approach by analyzing it from the perspective of gradient-bias injection [2]. As depicted in Figure 4, our method progressively introduces perturbations into the gradients calculated by the attacker. This process increases the optimization error that accumulates during the model-stealing process, rendering it more challenging for the attacker to successfully

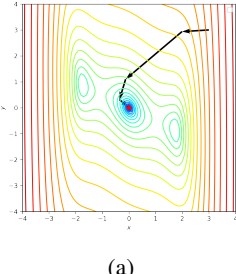
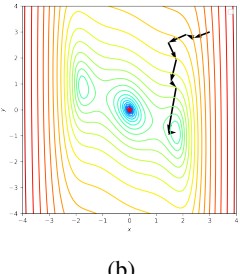

|(a)|(b)|

Figure 4: Theoretical explanations illustration: (a) We showcase the optimization trajectory employing the ground truth loss function that the attacker should ideally adopt (it remains inaccessible to the attacker owing to the unknown input perturbation generator). The cloned model converges toward the globally optimal stolen model, effectively emulating the target victim model. (b) Contrastingly, we depict the optimization trajectory utilizing a noisy and imprecise loss function, which is the actual choice of the attacker. The introduction of gradient bias due to the stochastic input perturbation generator causes the cloned model to deviate from the optimal stolen model, leading to ineffective model theft.

extract the target model. For a more comprehensive understanding, we have provided detailed theoretical analysis in the Appendix due to space constraints.

## 5 Experiments

### 5.1 Experimental Setup

**Datasets.** We perform experiments on four standard datasets used in DFME literature, including MNIST, CIFAR10, CIFAR100 [25] and MiniImageNet [52] (100 classes).

**Baselines.** We compare SOTA DFME and defense baselines. **Attack Baselines**: (1) *decision-based* DFME methods: DFMS-HL [47]. We do not compare to ZSDB3KD [51] since it requires a much larger number of queries and is very slow. (2) *score-based* DFME methods: MAZE [21] and DFME [51]. **Defense Baselines**: we compare to: (1) *Undefended*: the target model without using any defense strategy; (2) Random Perturb (*RandP*) [38]: randomly perturb the output probabilities. (3) *P-poison* [38]; (4) GRAD [33]: gradient redirection defense. We set a large $l_1$ perturbation budget equal to 1.0 for those defense baselines in the experiments

Table 1: Clone model accuracy after applying defense methods on **CIFAR-10** and **CIFAR-100** with ResNet34-8x as the target model, which provides **soft label**

| Attack | Defense | CIFAR10 Clone Model Architecture | | |
|---|---|---|---|---|
| | | ResNet18-8X | MobileNetV2 | DenseNet121 |
| DFME | undefended ↓ | $87.36 \pm 0.78\%$ | $75.23 \pm 1.53\%$ | $73.89 \pm 1.29\%$ |
| | RandP ↓ | $84.28 \pm 1.37\%$ | $70.56 \pm 2.23\%$ | $70.03 \pm 2.38\%$ |
| | P-poison ↓ | $78.06 \pm 1.73\%$ | $66.32 \pm 1.36\%$ | $68.75 \pm 1.40\%$ |
| | GRAD ↓ | $79.33 \pm 1.68\%$ | $65.82 \pm 1.67\%$ | $69.06 \pm 1.57\%$ |
| | MeCo ↓ | $\mathbf{51.68 \pm 1.96\%}$ | $\mathbf{46.53 \pm 2.09\%}$ | $\mathbf{61.38 \pm 2.41\%}$ |
| MAZE | undefended ↓ | $45.17 \pm 0.73\%$ | $23.28 \pm 1.67\%$ | $20.03 \pm 1.79\%$ |
| | RandP ↓ | $28.76 \pm 2.38\%$ | $22.03 \pm 1.50\%$ | $18.79 \pm 1.38\%$ |
| | P-poison ↓ | $26.81 \pm 2.19\%$ | $20.89 \pm 1.58\%$ | $\mathbf{17.08 \pm 2.28\%}$ |
| | GRAD ↓ | $26.06 \pm 1.81\%$ | $21.18 \pm 1.58\%$ | $18.09 \pm 1.72\%$ |
| | MeCo ↓ | $\mathbf{21.89 \pm 2.07\%}$ | $\mathbf{18.75 \pm 2.11\%}$ | $17.95 \pm 1.46\%$ |

| Attack | Defense | CIFAR100 Clone Model Architecture | | |
|---|---|---|---|---|
| | | ResNet18-8X | MobileNetV2 | DenseNet121 |
| DFME | undefended ↓ | $58.72 \pm 2.82\%$ | $28.36 \pm 1.97\%$ | $27.28 \pm 2.08\%$ |
| | RandP ↓ | $41.69 \pm 2.91\%$ | $22.75 \pm 2.19\%$ | $23.61 \pm 2.70\%$ |
| | P-poison ↓ | $38.72 \pm 3.06\%$ | $20.87 \pm 2.61\%$ | $21.89 \pm 2.93\%$ |
| | GRAD ↓ | $39.07 \pm 2.72\%$ | $20.71 \pm 2.80\%$ | $22.08 \pm 2.78\%$ |
| | MeCo ↓ | $\mathbf{29.57 \pm 1.97\%}$ | $\mathbf{12.18 \pm 1.05\%}$ | $\mathbf{10.79 \pm 1.36\%}$ |
| MAZE | | – | – | – |

to generate strong defense. That is, $||\boldsymbol{y} - \hat{\boldsymbol{y}}||_1 \leq 1.0$; where $\boldsymbol{y}$ and $\hat{\boldsymbol{y}}$ are the unmodified/modified output probabilities, respectively. We put more baseline details in Appendix 10.

**Implementation Details.** The random perturbation generator is a two-block ResNet structure with a filter size of 64, which is a small network compared to the backbone (only accounts for a tiny proportion of the backbone). For decision-based DFME methods, following [47], we use a query budget of $10M$ for CIFAR100 and $8M$ for CIFAR-10. For score-based DFME methods, following [51], we set the number of queries to be $2M$ for MNIST, $20M$ for CIFAR10, and $200M$ for CIFAR100, respectively. We perform each experiment for 5 runs with a mean and standard deviation of results. We provide more implementation details in Appendix 7.

### 5.2 Results of Defense against DFME

**Clone model accuracy.** For score-based DFME setting (soft label), we show the results on CIFAR10 and CIFAR100 in Table 1. For decision-based DFME setting (hard label), we show the results in

Table 2: Clone model accuracy after applying different defense methods on **CIFAR-10** and **CIFAR-100** with ResNet34-8x as the target model, which only provides **hard label**

| Attack | Defense | **CIFAR10** Clone Model Architecture | | | **CIFAR100** Clone Model Architecture | | |
|---|---|---|---|---|---|---|---|
| | | ResNet18-8X | MobileNetV2 | DenseNet121 | ResNet18-8X | MobileNetV2 | DenseNet121 |
| DFMS-HL | undefended ↓ | $84.67 \pm 1.90\%$ | $79.28 \pm 1.87\%$ | $68.87 \pm 2.08\%$ | $72.57 \pm 1.28\%$ | $62.71 \pm 1.68\%$ | $63.58 \pm 1.79\%$ |
| | RandP ↓ | $84.02 \pm 2.31\%$ | $78.71 \pm 1.93\%$ | $68.16 \pm 2.23\%$ | $72.43 \pm 1.43\%$ | $62.06 \pm 1.82\%$ | $63.16 \pm 1.73\%$ |
| | P-poison ↓ | $84.06 \pm 1.87\%$ | $79.02 \pm 1.96\%$ | $68.05 \pm 2.17\%$ | $71.83 \pm 1.32\%$ | $61.83 \pm 1.79\%$ | $62.73 \pm 1.91\%$ |
| | GRAD ↓ | $84.28 \pm 1.95\%$ | $78.83 \pm 1.91\%$ | $68.11 \pm 1.93\%$ | $71.89 \pm 1.37\%$ | $62.60 \pm 1.71\%$ | $62.57 \pm 1.80\%$ |
| | MeCo ↓ | $\mathbf{76.86 \pm 2.09\%}$ | $\mathbf{71.22 \pm 1.87\%}$ | $\mathbf{62.33 \pm 2.01\%}$ | $\mathbf{59.30 \pm 1.70\%}$ | $\mathbf{55.32 \pm 1.65\%}$ | $\mathbf{56.80 \pm 1.86\%}$ |

Table 3: Target model utility (test accuracy and $l_1$ norm of the output probabilities perturbation magnitude)

| Method | MNIST | | CIFAR10 | | CIFAR100 | |
|---|---|---|---|---|---|---|
| | ACC ↑ | $l_1$ norm ↓ | ACC↑ | $l_1$ norm ↓ | ACC↑ | $l_1$ norm ↓ |
| undefended | $98.91 \pm 0.16\%$ | 0.0 | $94.91 \pm 0.37\%$ | 0.0 | $76.71 \pm 1.25\%$ | 0.0 |
| RandP | $98.52 \pm 0.19\%$ | 1.0 | $93.98 \pm 0.28\%$ | 1.0 | $75.23 \pm 1.39\%$ | 1.0 |
| P-poison | $\mathbf{98.87 \pm 0.35\%}$ | 1.0 | $94.58 \pm 0.61\%$ | 1.0 | $75.42 \pm 1.21\%$ | 1.0 |
| GRAD | $98.73 \pm 0.31\%$ | 1.0 | $\mathbf{94.65 \pm 0.67\%}$ | 1.0 | $\mathbf{75.60 \pm 1.45\%}$ | 1.0 |
| MeCo | $98.63 \pm 0.28\%$ | $\mathbf{0.0126}$ | $94.17 \pm 0.56\%$ | $\mathbf{0.099}$ | $75.36 \pm 0.68\%$ | $\mathbf{0.312}$ |

Table 2. Due to the space limitations, we present the results on *MiniImageNet and MNIST* in Table 8 and 7 in Appendix. ↓ indicates the accuracy the lower, the better; ↑ indicates the accuracy the higher, the better. For CIFAR10/CIFAR100 in Table 1, we use ResNet34 [13] as the target model. The clone model architecture includes, ResNet-18, MobileNetV2 [46], DenseNet121 [15]. We further change the target model architecture as GoogLeNet [48] for CIFAR10 in Table 9 in Appendix.

The results show MeCo significantly reduces the effectiveness of existing DFME methods by up to 35% and is substantially more effective than the compared methods since (1) RandP lacks data-dependent information; it maintains the utility for almost all the query data, which is unnecessary for attack queries since the attacker can still learn useful information. (2) P-poison needs a random initialized surrogate attacker model, which acts as an adversary model. (3) GRAD needs to know the attack query set to train the surrogate model. Those surrogates have large gaps compared to the DFME attacker model. Those methods thus perform poorly since the attack query data distribution and model are unknown to the defender. While MeCo does not need a surrogate model.

**Target model utility.** We evaluate the target model utility by (1) target model test accuracy after adopting the defense strategy; (2) $l_1$ norm between the target model output probabilities with and without input perturbation averaged on the test dataset, i.e., $\mathbb{E}_{\boldsymbol{x} \sim \mathcal{D}_{test}} ||T(\boldsymbol{x} + h_{\boldsymbol{\omega}}(\boldsymbol{x}, \boldsymbol{\epsilon})) - T(\boldsymbol{x})||_1$. The results (see Table 3 (MNIST with LeNet, CIFAR10 and CIFAR100 with ResNet34-8x)) indicate that MeCo maintains the target model utility with a slight trade-off of benign accuracy but with much better preservation of output probabilities in terms of $l_1$ norm. The baselines have large perturbations because the baseline defense methods perturb all the query data with the same magnitude ($l_1$ perturbation budget of 1.0); this is unnecessary since the in-distribution benign query does not need such significant perturbation. In contrast, MeCo explicitly optimizes the $l_1$ norm of perturbation with DRO on the simulated test data distribution so that the perturbation magnitude is much smaller.

### 5.3 Application on Data-Based ME (DBME) Attack

We apply MeCo to defend against traditional DBME methods with Knockoff Nets [37] and Jacobian-Based Dataset Augmentation (JBDA) [41]. We present the results in Appendix 9.1. MeCo can significantly outperform existing methods and further reduce the clone model accuracy. When the distribution of attack query data is closer to the training data distribution of the target model, the effectiveness of MeCo defense weakens. This is due to the fact that MeCo applies smaller perturbations to the data distribution similar to the training data of the target model as a result of defensive training.

### 5.4 Adaptive Attacks

We further analyze the robustness of MeCo against the attacker's countermeasures. Namely, we consider the situations where attackers know about our defense and have tailored the attacks to our defense method by adding data-dependent random perturbation to the query inputs to learn an additional random perturbation generator. In Table 4, MeCo is still effective since the random perturbation added by the defender and attacker are different due to randomness. There is still a mismatch in the KD loss and zeroth-order gradient estimation for the attacker. Interestingly, the performance of the clone model becomes even worse after the attacker performs an adaptive attack.

We believe this is because after the attacker adds the random perturbation, they will need to learn a random function, increasing the difficulty of model extraction. In addition, for score-based DFME, following [38, 33], we also compare to the adaptive attack method where attackers use only hard label, not probability outputs. We show the results in Table 5. MeCo is still very effective since the pseudo samples are nearer to the decision boundary, and perturbing the input would easily change the model outputs on those samples. Consequently, the attacker still cannot learn useful information.

## 5.5 Ablation Study

**Effect of DRO.** We evaluate how much improvement DRO can bring to the model utility preservation in Table 12 in Appendix. We can observe that with DRO, our method significantly improves the model utility by $5.3\% - 5.6\%$ compared to the one without DRO training.

**Effect of different query budgets for attacker.** To evaluate the effect of different query budgets on defense performance, we evaluate the clone model accuracy with varying defense methods on CIFAR10 in Figure 5 in Appendix. MeCo substantially outperforms various baselines with varying query budgets.

Table 4: Clone model accuracy after applying **adaptive attack** on **CIFAR10** with ResNet34-8x as the target model

| Attack | Defense | Clone Model Architecture | | |
|--------|---------|-----------------|-------------|-------------|
| | | ResNet18-8X | MobileNetV2 | DenseNet121 |
| DFME | undefended ↓ | $87.36 \pm 0.78\%$ | $75.23 \pm 1.53\%$ | $73.89 \pm 1.29\%$ |
| | MeCo ↓ | $51.68 \pm 1.96\%$ | $46.53 \pm 2.09\%$ | $61.38 \pm 2.41\%$ |
| | MeCo, adaptive ↓ | $\mathbf{25.79 \pm 0.81\%}$ | $\mathbf{20.18 \pm 1.17\%}$ | $\mathbf{22.32 \pm 1.82\%}$ |
| MAZE | undefended ↓ | $45.17 \pm 0.73\%$ | $23.28 \pm 1.67\%$ | $20.03 \pm 1.79\%$ |
| | MeCo ↓ | $21.89 \pm 2.07\%$ | $18.75 \pm 2.11\%$ | $\mathbf{16.31 \pm 1.76\%}$ |
| | MeCo adaptive ↓ | $\mathbf{19.82 \pm 2.03\%}$ | $18.27 \pm 2.23\%$ | $17.08 \pm 2.28\%$ |

Table 5: Clone model accuracy that attacker only uses **hard label** instead of output probabilities on **CIFAR10**

| Attack | Defense | Clone Model Architecture | | |
|--------|---------|-----------------|-------------|-------------|
| | | ResNet18-8X | MobileNetV2 | DenseNet121 |
| DFME | undefended ↓ | $87.36 \pm 0.78\%$ | $75.23 \pm 1.53\%$ | $73.89 \pm 1.29\%$ |
| | MeCo, soft label ↓ | $51.68 \pm 1.96\%$ | $46.53 \pm 2.09\%$ | $61.38 \pm 2.41\%$ |
| | MeCo, hard label ↓ | $\mathbf{41.23 \pm 0.58\%}$ | $\mathbf{38.67 \pm 0.97\%}$ | $\mathbf{42.31 \pm 1.78\%}$ |

**Hyperparameter Sensitivity.** We evaluate the hyperparameter sensitivity for $\gamma$ and $Q$ in Table 10 and Table 11 in Appendix. We observe that model utility increases as $\gamma$ increases with a trade-off of a decrease in defense performance. Results show that $Q = 2$ performs the best. With the increase of $Q$, the simulated test data may be harder to learn; thus, the benign accuracy slightly decreases.

**Test time speed and memory comparisons.** We compare the running time of best-performing methods in Table 13 in Appendix. MeCo achieves $17 \sim 172 \times$ speed up on CIFAR10 and CIFAR100. This is because P-poison and GRAD solve computationally expensive optimization during testing. In contrast, MeCo does not need this optimization. We provide the results in Table 14 in Appendix for memory consumption evaluation. MeCo is very memory efficient compared to baselines.

**Training efficiency.** We compare our training efficiency to baselines in Table 15 in Appendix. MeCo increases the training cost by $1.3 \times$. However, MeCo substantially improves the test time running efficiency. We believe this slightly additional computation cost is worth it.

## 6 Conclusion

In this paper, we explore the defense against DFME. We propose a memory and computation efficient (MeCo) defense method through distributionally robust defensive training by adding a data-dependent random perturbation generator to perturb the input data so that the attacker cannot steal useful information from the black-box model. At the same time, MeCo maintains the target model utility. Extensive experiments demonstrate MeCo's effectiveness, computation, and memory efficiency.

**Limitations and Broader Impacts**. Our proposed defensive training helps build safe and trustworthy AI systems. This would be beneficial for protecting current large-scale pre-trained models used in the public APIs since they spend lots of money and time to tune and deploy those valuable large-scale models. However, sharing models and insights is a key driver of progress in the AI research field, and if the community becomes more protective of their models, it could slow down advancements.

**Acknowledgement** This research was partly supported by NSF through grants IIS-1910492 and IIS-2239537. Tongliang Liu was partially supported by the following Australian Research Council projects: FT220100318, DP220102121, LP220100527, LP220200949, and IC190100031. We also gratefully acknowledge the support of NVIDIA Corporation with the donation of the A5000 used for this research.

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
