# OpenReview forum: "Defending against Data-Free Model Extraction by  Distributionally Robust Defensive Training"
_NeurIPS.cc/2023/Conference — NeurIPS 2023 poster_

### Official Review · Reviewer_xYzi · 2023-07-02

**Soundness:** 3 good
**Presentation:** 3 good
**Contribution:** 3 good
**Rating:** 5
**Confidence:** 3

**Summary:**

This paper presents a new method to defend against depth function preserving model extraction (DFME) attacks. The method proposed by the authors, called MeCo, adds data-dependent random perturbations to the input data, making it difficult for attackers to extract useful information from the black-box model. The authors claim that MeCo effectively reduces the accuracy of the cloned model while maintaining the utility of the target model. The approach was evaluated on several data sets and compared to existing defense strategies, showing excellent performance in terms of effectiveness, computational efficiency, and memory usage.

**Strengths:**

1.The quality of the work is evident in the extensive experiments conducted and the clear presentation of results.
2.The main advantage of the paper is its novel approach to defend against DFME attacks. The proposed method, MeCo, is both effective and efficient, which is demonstrated by experimental results. However, these advantages may be affected by a lack of theoretical support.

**Weaknesses:**

1.The paper only conducted experiments in cifar10, cifar100 and mnist data sets, but the lack of relevant experiments on large data sets led to some doubts about the extensibility of the model.
2.The model may need more testing and proof in generalization.

**Questions:**

Can you provide more details on the theoretical basis of the proposed approach?
How does the proposed method perform under different attack scenarios?

**Limitations:**

The authors acknowledge the lack of theoretical analysis as a limitation of the current work and take it as a direction for future work. They also mention the potential benefits of their proposed defense training for protecting large-scale pre-trained models used in public apis. However, a more detailed discussion of the possible negative social impacts of the proposed approach would be useful.

---

> ### Author Rebuttal · Authors · 2023-08-09
>
> We would like to express our sincere gratitude for your constructive feedback.
>
> **Q1**:  lack of relevant experiments on large data sets led to some doubts about the extensibility of the model.
>
> **A**: Thanks for your suggestions. As requested, we included the results obtained on the MiniImageNet [1] dataset with 100 classes, which is a subset of the ImageNet dataset. The results are presented in Table 8 of the Appendix in our submission.  For convenience, we also present the results on MiniImageNet  in the following table.
>
> In addition, we took into consideration the dataset evaluated in the current state-of-the-art model extraction defense methods, as referenced in [2]. As a result, we extended our experiments to include the CUB200 dataset [3] with 200 image classes, which, as mentioned in [2], represents the largest dataset utilized for model extraction defense. The detailed outcomes of our experiments on the CUB200 dataset are provided in the following tables.  We hope that these additional results further contribute to the comprehensive evaluation of our proposed model and its defense against model extraction techniques.
>
> *Clone Model Accuracy with Different Defense Methods on **MiniImageNet**  with Different Clone Model Architecture*
>
> | Attack | Defense  | ResNet18-8X | MobileNetV2 | DenseNet121 |
> |--------|----------|------------|-------------|-------------|
> |        | Undefended | 35.89% ± 3.97% | 28.71% ± 3.25% | 25.05% ± 3.68% |
> | DFME   | RandP | 30.76% ± 4.09% | 22.06% ± 3.83% | 20.23% ± 3.97% |
> |  (Soft Label)   | P-poison | 29.36% ± 4.23% | 21.83% ± 3.77% | 20.01% ± 3.89% |
> | | GRAD | 29.87% ± 3.76% | 21.65% ± 3.75% | 19.82% ± 3.77% |
> |        | MeCo | **23.29% ± 3.83%** | **17.83% ± 3.67%** | **16.73% ± 3.88%** |
> |        |          |            |             |             |
> |        | Undefended | 46.72% ± 4.86% | 40.35% ± 4.97% | 38.71% ± 3.85% |
> | DFMS-HL | RandP | 45.09% ± 4.93% | 39.51% ± 4.83% | 38.08% ± 3.95% |
> | (Hard Label) | P-poison | 45.16% ± 5.03% | 39.06% ± 4.72% | 37.78% ± 4.26% |
> |        | GRAD | 45.32% ± 5.21% | 39.17% ± 4.85% | 37.85% ± 4.32% |
> |        | MeCo | **39.23% ± 4.83%** | **35.81% ± 4.69%** | **32.30% ± 4.56%** |
>
> *Clone Model Accuracy with Different Defense Methods on **CUB200**  with Different Clone Model Architecture*
>
> | Attack | Defense  | ResNet18-8X | MobileNetV2 | DenseNet121 |
> |--------|----------|------------|-------------|-------------|
> |        | Undefended | 49.75% ± 3.82% | 32.83% ± 3.53% | 29.08% ± 3.92% |
> | DFME   | RandP | 38.89% ± 4.16% | 26.85% ± 3.96% | 23.81% ± 4.05% |
> |        | P-poison | 36.45% ± 4.38% | 25.31% ± 3.91% | 22.09% ± 4.03% |
> | (Soft Label) | GRAD | 31.70% ± 3.61% | 24.78% ± 3.89% | 21.91% ± 3.86% |
> |        | MeCo | **20.78% ± 4.67%** | **18.65% ± 4.90%** | **15.86% ± 4.51%** |
> |        |          |            |             |             |
> |        | Undefended | 58.89% ± 5.03% | 49.35% ± 5.18% | 46.71% ± 4.37% |
> | DFMS-HL | RandP | 55.28% ± 5.39% | 46.67% ± 5.65% | 45.19% ± 4.32% |
> | (Hard Label) | P-poison | 52.71% ± 5.36% | 45.28% ± 5.31% | 43.76% ± 4.91% |
> |        | GRAD | 51.43% ± 5.73% | 46.51% ± 5.50% | 42.39% ± 4.82% |
> |        | MeCo | **43.31% ± 5.16%** | **38.27% ± 5.03%** | **33.42% ± 4.95%** |
>
> Reference:
>
> [1] Matching Networks for One Shot Learning.  NeurIPS 2016.
>
> [2] How to Steer Your Adversary: Targeted and Efficient Model Stealing Defenses with Gradient Redirection. ICML 2022.
>
> [3] Caltech-UCSD Birds 200. 2010
>
> **Q2**: The model may need more testing and proof in generalization.
>
> **A**: Thank you for your suggestions. In response to your concerns, we made significant enhancements to our work based on your feedback. Specifically, we have included additional experimental results and in-depth theoretical explanations for the proposed methods.
>
> In response to your first question **Q1**, we have incorporated supplementary experimental results that shed further light on the performance and generalization of our approach. These results can be found in **Q1**.
>
> Furthermore, we have provided a comprehensive theoretical basis for our proposed methods. Our aim was to elucidate the underlying principles and mechanisms that support the efficacy of our approach. These theoretical explanations are also elaborated upon in the global response. We believe that these additional elements substantially strengthen the quality and comprehensiveness of our work.
>
> **Q3**: Can you provide more details on the theoretical basis of the proposed approach? How does the proposed method perform under different attack scenarios?
>
> **A**:  Thank you for your comments, we provided a comprehensive theoretical basis for our proposed methods. Our aim was to elucidate the underlying principles and mechanisms that support the efficacy of our approach. These theoretical explanations are also elaborated upon **in the global response**.  Furthermore, we supplemented our theoretical explanations with visualization, which can be found in Figure 1 within the **'rebuttal.pdf' document as part of the global response**.
>
> Furthermore, our method can also be viewed as random smoothing: $T_{\theta_T}^{S}(x) = \int T_{\theta_T}(x + h_{\omega}(x, \epsilon)) d\epsilon$, where $\epsilon \sim N(0, I)$. Compared to learning a single function $T_{\theta_T}(x)$, learning the smoothed function $T_{\theta_T}^{S}(x)$ necessitates a larger number of queries for attacker due to the requirement of calculating the integral. By necessitating a larger query budget, our approach makes it considerably harder for attackers to successfully clone the model.
>
> **Q4**:  Negative Social Impact
>
> **A**:   Sharing models and insights is a key driver of progress in the AI research field, and if the community becomes more protective of their models, it could slow down advancements.
> Stricter model extraction defenses could create economic barriers for startups that rely on reverse engineering or analyzing existing models as part of their business strategy.

---

### Official Review · Reviewer_e67e · 2023-07-05

**Soundness:** 3 good
**Presentation:** 3 good
**Contribution:** 3 good
**Rating:** 6
**Confidence:** 4

**Summary:**

* __Problem Statement__: The paper proposes a defense against data-free model extraction attacks (where the attacker black-box queries a victim image classifier, s.t queries are then used to train a clone model)
* __Approach__: The proposed defense "DRO" returns class probabilities as a result of the defender *perturbing* the input image. The objective of the perturbation is to flip the top-1 predicted class of attacker's queries (characterized by lying close to decision boundaries), but while preserving utility (measured by L1 distance to unperturbed probabilities).
* __Experimental findings__: The approach is evaluated on 4x standard datasets, compared against 3x recent baseline approaches and in addition provides more studies (e.g., ablation). Results indicate that the proposed approach outperforms the baselines (in terms of both accuracy and utility).

**Strengths:**

1. __Originality__: 4/5. The approach is original and well-motivated. Although some ideas have been explored before (i.e., perturbing outputs, changing gradient direction), the proposed approach explores a novel objective (learnt perturbation of inputs to introduce a targeted shift in prediction) and proposes a reasonable way to achieve it.
2. __Quality__: 4/5. The paper is high-quality. The insights motivating the approach are sound and the experimental section is exhaustive.
3. __Clarity__: 3/5. The approach is somewhat clear (certain technical choices/notation are unclear; more under questions).
4. __Significance__: 3/5. The approach is somewhat significant -- it clearly improves upon existing defenses, however against a very specific class of model extract attacks. Specifically, "data-free" attacks that have shown to be successful largely in extreme settings (such as with attacker querying the victim millions of times).

**Weaknesses:**

1. __DRO optimization setup - unclear__: I found the intuition for the defense objective (Fig 2, L148-207) clear. However, it is unclear how the objective formulation (Eq. 4-6) achieves this. Based on Fig. 2 and L183-190, it appears that the requirement is to perturb images s.t the label is flipped (i.e., maximize CE loss wrt parameters $\omega$), however the outer maximization in Eq. 6 seems to do the opposite? Furthermore, in Eq. 6, since the utility is upper-bounded $||\cdots||_1 \le \tau$, it appears that the perturbation generator $h_\omega$ can simply learn an identity function?
2. __Evaluation curves missing__: The paper although extensively evaluates the defense, it appears that it is primarily for a specific defense effectiveness-utility operating point (specific value of $\gamma$ and $Q$). This contrasts prior works which evaluates a curve of attacker's vs. defender's classification accuracies (e.g., Fig. 4 GRAD, Fig. 4 PredPoison). Without such a curve, it's somewhat unclear to determine the overall performance of the attack.
3. __Perturbation budget $\le$ 1.0__: The L_1 perturbation budget $\tau$ in the defense chosen appears to be a fixed to 1 (L222, L298) seems unreasonably high. After all, this allows the defender to always flip the top-1 class-label.

**Questions:**

1. Hyperpameter choices
    1. How were the defense hyperparameters determined? I'm concerned that if they are a result of hyperparam searches (e.g., over a sweep of attacks), it might not generalize to novel attacks.
    2. Hyperparameter choices: Are the defense hyperparameters fixed for all choices of datasets, clone model architectures, and attacks?
2. DRO optimization objective vs. Adversarial training: Can the authors clarify on how the min-max optimization objective (Eq. 4) is different from adversarial training? After all, in both cases, the objective is to find parameters $\theta$ s.t predictions are robust to perturbed inputs.
3. (Suggestion) Sec 5.4 Adaptive attacker: Please elaborate on how the attacker "adapts". It is unclear what the attacker does from L341.

**Limitations:**

Limitations are adequately discussed in Sec. 6.

---

> ### Author Rebuttal · Authors · 2023-08-09
>
> We would like to express our sincere gratitude for your insightful comments.
>
> **Q1.1**: DRO optimization setup: Based on Fig. 2 and L183-190, it appears that the requirement is to perturb images s.t the label is flipped (i.e., maximize CE loss wrt parameters )?
>
> **A1.1**:  Eq 4-6 is to **maintain the model utility on benign inputs**.  Given that perturbations applied to benign inputs can diminish model utility by altering probability outputs, the rationale behind this Distributionally Robust Optimization (DRO) formulation is to ensure the robust generalization of the model on perturbed test data. Consequently, the objective of this optimization is to minimize the loss function across the simulated test set from the training data.
>
> **Q1.2**: DRO optimization setup: in Eq. 6, since the utility is upper-bounded, it appears that the perturbation generator can simply learn an identity function?
>
> **A1.2**:  Eq.6 is a constraint (not optimization objective) for the optimization problem in Eq. 4.  Learning an identity function is only  *a feasible solution of the optimization, but is not the optimal solution*. This is due to the interdependence between the parameters of the perturbation generator and parameters of the target model.
>
> **Q2**: evaluate a curve of attacker's vs. defender's classification accuracies.
>
> **A**:   Thanks for your question, please refer to **Figure 2 in rebuttal.pdf in the global response** for the evaluation curves.
>
> **Q3**:  The L_1 perturbation budget appears to be fixed to 1 seems unreasonably high.
>
> **A**:  Thanks for pointing out this.
> **Set a perturbation budget equal to 1 is intended to create  stronger baseline defense methods for comparisons**. A larger perturbation budget can **strengthen the defense of the compared baseline methods** by allowing more flexibility to adjust the prediction logits. To further illustrate the impact of varying perturbation budgets on defense methods, we have included additional results with smaller perturbation budgets of 0.2 and 0.5 in the following table. The results show that the defense performance of compared baseline defense methods appears weaker when the defender's perturbation budget is constrained. In such cases, the defender can only perturb the output probabilities using a smaller budget, which poses a challenge to existing defense mechanisms.
>
> *Perturbation Budget of 0.2*
>
> |Attack| Defense|ResNet18-8X|MobileNetV2|DenseNet121|
> |--------|----------|------------|-------------|-------------|
> ||Undefended|87.36%±0.78%|75.23%±1.53%|73.89%±1.29%|
> |DFME|RandP|86.32%±1.05%|74.67%±2.02%|72.16%±2.05%|
> ||P-poison|83.17%±1.66%|73.32%±1.45%|71.68%±1.51%|
> ||GRAD|84.26%±1.72%|71.82%±1.67%|71.23%±1.68%|
> ||MeCo|**51.68%±1.96%**|**46.53%±2.09%**|**61.38%±2.41%**|
>
> *Perturbation Budget of 0.5*
>
> |Attack|Defense|ResNet18-8X|MobileNetV2|DenseNet121|
> |--------|----------|------------|-------------|-------------|
> ||Undefended|87.36%±0.78%|75.23%±1.53%|73.89%±1.29%|
> |DFME|RandP|85.53%±1.65%|74.62%±2.37%|71.56%±2.79%|
> ||P-poison|81.19%±1.86%|69.71%±1.56%|70.93%±1.58%|
> ||GRAD|82.89%±1.82%|68.30%±1.75%|71.32%±1.73%|
> ||MeCo|**51.68%±1.96%**|**46.53%±2.09%**|**61.38%±2.41%**|
>
> **Q4**: How were the defense hyperparameters determined? Are the defense hyperparameters fixed for all choices of datasets, clone model architectures, and attacks?
>
> **A**:  Thank you for your questions. In the data-free model extraction setting, where the attack query data distribution is unknown, we use a surrogate dataset to represent the attack query data and then choose the hyperparameters that achieve the worst performance on the validation dataset of target dataset. For example, we adopt CIFAR100 dataset as a surrogate query dataset to query the target victim model trained on CIFAR10. Our next step involves defensive training using our proposed method on CIFAR10 training dataset. To evaluate the effectiveness of our defense, we simulate the model stealing process by calculating the loss function of the attacker. This is done by comparing the outputs of the cloned model and the target model on CIFAR100 dataset.
> We choose the hyperparameters that lead to the worst performance on the validation set of CIFAR10 with the extracted model.
>
> These selected hyperparameters should be tuned for each target dataset, i.e., dataset-dependent. After tuning for each dataset,  then the hyperparameters remain fixed across clone model architectures, and various attacks.
>
> **Q5**: DRO optimization vs. Adversarial training
>
> **A**: Thanks for your question. We'd like to clarify the distinction between adversarial training and distributionally robust optimization (DRO).
> Adversarial training focuses on optimizing the model's performance against worst-case perturbations for **each individual training data point**. The goal is to enhance the model's robustness to small adversarial perturbations. However, this approach will lead to a drop in performance on the test data, as the model becomes *too conservative*.
>
> On the other hand, DRO aims to optimize the model's performance by considering the worst-case performance over **a set of possible data distributions**. Instead of focusing solely on individual data points, DRO accounts for *uncertainties in the data and generalizes well to different distributions*. As a result, DRO tends to improve performance on the test data, striking a balance between standard empirical optimization and adversarial training.
>
> **Q6**:  Adaptive attacker: Please elaborate on how the attacker "adapts"
>
> **A**: The adaptive strategy involves introducing data-dependent perturbations to the clone model's input to simulate the defense mechanism. This approach aims to bridge the gap between the defense used during testing and the training of the clone model for attacker. By incorporating the defense mechanism into the clone model's input, attacker seeks to better adapt to defender's protection strategy. This allows attacker to improve the effectiveness of their attacks.

---

### Official Review · Reviewer_v48m · 2023-07-06

**Soundness:** 3 good
**Presentation:** 3 good
**Contribution:** 3 good
**Rating:** 5
**Confidence:** 3

**Summary:**

In this paper, authors propose a novel principled defensive training framework that substantially improves the memory and computation efficiency during deployment to defend against DFME attacks and a distributionally robust optimization method to randomly perturb the inputs to defend against DFME effectively while maintaining the model utility simultaneously.

**Strengths:**

1. I agree that this work would be beneficial for protecting current large-scale pre-trained models used in the public APIs.

2. The experiment in this article is sufficient and the description is clear. And the method proposed by the author is novel to a certain extent.

3. The author's description of the existing research background is clear, and the analysis of the problems existing in the field is satisfactory.

**Weaknesses:**

1. In my opinion, the method proposed by the author has defects in protecting the performance of the original model, that is, it does not fully take into account the impact on the performance of the target model, which can also be seen from Table 3. After all, the weak accuracy improvement of these models in practical applications is expensive.
2. As stated by the author, the methodology of this article lacks a theoretical explanation. The intuitive explanation of the mechanism of action of the model is not deep enough.

**Questions:**

See weakness

**Limitations:**

This method does not adequately evaluate the negative effects of the target model.

---

> ### Author Rebuttal · Authors · 2023-08-09
>
> We would like to express our appreciation for your valuable suggestions.
>
>
> **Q1**: In my opinion, the method proposed by the author has defects in protecting the performance of the original model, that is, it does not fully take into account the impact on the performance of the target model, which can also be seen from Table 3. After all, the weak accuracy improvement of these models in practical applications is expensive.
>
> **A**:  We appreciate your comments and your acknowledgment of the challenges involved in defending against model extraction attacks. In practice, developing effective defenses necessitates striking a balance between benign accuracy and defense performance.
>
>   * In our research, we thoroughly evaluated various defense baselines, including GRAD, P-poison, and Random defense, etc. It is important to note that most compared baselines sacrifice benign accuracy to enhance defense performance, our proposed defense method strives to achieve a more optimal trade-off.
>
> * As demonstrated in Table 3, our defense methods significantly reduce output probability perturbations when compared to other defense approaches. This improvement highlights the effectiveness of our method in mitigating the risk of model extraction attacks with only slightly compromising the model's performance on benign inputs.  This is beneficial for benign users in practical applications, while the compared defense methods need to significantly modify the output probabilities.
>
> * Regarding the slight reduction in benign accuracy, it is essential to emphasize that this decrease is minimal and does not significantly impact the overall model performance. In most cases, our proposed method only slightly reduces the benign accuracy, e.g., 0.28% on Minist, 0.74% on CIFAR10, and 1.35% on CIFAR100. Despite this marginal drop in accuracy, our method remains competitive with the baseline defense methods. Additionally, our method is significantly more efficient and effective in defending against model extraction attacks as shown in other tables.
>
>  * Overall, our work aims to strike a balance between robustness against model extraction attacks and maintaining competitive accuracy levels.
>
>
>
> **Q2**: As stated by the author, the methodology of this article lacks a theoretical explanation. The intuitive explanation of the mechanism of action of the model is not deep enough.
>
> **A**: Thank you for pointing out this. We provided a comprehensive theoretical foundation for our proposed methods, with the intention of illuminating the fundamental principles and mechanisms that underpin the effectiveness of our approach. These theoretical elucidations are further illustrated in the **global response**.  Furthermore, we also included figure visualizations to complement our theoretical explanations, as depicted in **Figure 1 in the 'rebuttal.pdf' within the global response**.
>
>
> Our method can also be analyzed from another perspective. The introduction of random input perturbation allows our method to act as a form of random smoothing, defined by the function: $T_{\theta_T}^{S}(x) = \int T_{\theta_T}(x + h_{\omega}(x, \epsilon)) d\epsilon$, where $\epsilon \sim N(0, I)$. This smoothing technique enhances our method's resilience against model extraction attacks. It is important to note that compared to learning a single function $T_{\theta_T}(x)$, the process of learning the smoothed function $T_{\theta_T}^{S}(x)$ necessitates a larger number of queries for the attacker due to the computational complexity of calculating the integral.
> By necessitating a larger query budget, our approach makes it considerably harder and more resource-intensive for attackers to successfully clone the model. As a result, our defense method offers a robust and effective barrier against model replication.

---

> > ### Comment · Reviewer_v48m · 2023-08-17
> > **Thanks for the responses**
> >
> > Thanks for the responses. I don't have any other questions anymore.

---

> > > ### Author Response · Authors · 2023-08-17
> > > **Thanks for your confirmation**
> > >
> > > Thank you for your acknowledgement.

---

### Official Review · Reviewer_VEKG · 2023-07-07

**Soundness:** 2 fair
**Presentation:** 3 good
**Contribution:** 2 fair
**Rating:** 5
**Confidence:** 4

**Summary:**

This paper proposes a defense against data-free model extraction attack (DFME). The basic idea is to add randomized data-dependent perturbations to the input query. It proposes a new training method that considers the perturbation generator to mitigate the risks of dropping the benign accuracy. Namely, it will trian the model and the perturbation generator so that the model can behave properly on the benign queries.


**Strengths:**

It is interesting to add random perturbations to the test data for defending against the DFME attacks.

The paper is well organized, with all items discussed.


**Weaknesses:**

From the very high level, I do not see how this works. The method tries to add perturbations to the test data. This will obviously decrease the benign accuracy. But I am not able to identify what makes the training method work. In DFME attacks, the queries are just normal queries, which can be similar to the ones used in training or testing. I am not sure how the method can work without identifying different types of queries.

One argument may be the query in the attack will be different from the ones used in training. Does that mean that the defense assumes the attack query out of distribution? I think out-of-distribution attack queries will surely degrade the attack performance.

Is there a theoretical guarantee of the proposed method?


**Questions:**

How does the method tell the differences between normal and malicious queries?

Is there a theoretical guarantee of the proposed method?


**Limitations:**

The paper discussed the limitations and broader impacts.

---

> ### Author Rebuttal · Authors · 2023-08-09
>
> We sincerely appreciate your thoughtful feedback.
>
> **Q 1.1**: Adding perturbation to test data decreases benign accuracy.
>
> **A 1.1**: We explain why our method maintains benign accuracy by considering three cases: (We invite you to refer to **Figure 3 for visual depiction in the "rebuttal.pdf" within the global response**)
>
> * In the first scenario, target model undergoes standard training on the target dataset. Although model utility remains intact, its vulnerability to model theft is heightened due to the lack of defense mechanisms.
>
> * Second, random input perturbations are introduced to query data without our proposed defensive training. As a result, target victim model gains protection against theft attempts, but this comes at the cost of compromised model utility.
>
> * To tackle these challenges and achieve the goals of preserving model utility and establishing robust defenses against model extraction attacks, we introduce our distributionally robust optimization approach (outlined in Equations 4-6 in the main text). This procedure aims to uphold model utility by optimizing worst-case generalization on perturbed test data, achieved by simulating testing data from the training data. This approach applies the common assumption that benign queries share the same distribution as training data, a prevalent assumption in existing model extraction defenses [5].
>
> **Q 1.2**: DFME queries can be similar to the ones in training or testing. How does the method tell the differences between normal and malicious queries?
>
> **A 1.2**:
> * First, we would like to clarify that DFME attacks commonly operate under the assumption that the attacker lacks knowledge of the training data distribution employed by the target model. In other words, the attacker operates in a "data-free" manner, being able to employ **flexible data distributions that differ from the training data distribution** to interact with the target model [3,4].
> Furthermore, in light of the insights and findings presented in PoW [1], the data utilized for attack queries is synthetic and out-of-distribution (OOD) since these data are generated synthetically using a data generator. Consequently, they possess a **different distribution** compared to the data encountered during training or testing.
>
> * Additionally, we leverage the insights in [2], which observe the fact that attack queries tend to reside closer to decision boundary, while normal queries are positioned further away. Consequently, the output probabilities associated with attack queries experience more significant perturbations upon the application of input perturbation. Conversely, owing to normal queries distance from the decision boundary, the output probabilities of normal queries undergo comparatively milder perturbations.
>
> * We employ distributionally robust optimization, as Equations 4-6 in the main text, to ensure the preservation of model utility on the test data through training a data-dependent perturbation generator, denoted as $h_{\omega}(x, \epsilon)$. When $x$ corresponds to a benign query, one that aligns with the distribution of training data, the perturbation magnitude is inherently restrained due to the defensive training of the perturbation generator. This restraint arises from the training and simulation of these data points derived from training data, thereby guaranteeing model utility. In contrast, when $x$ pertains to an attack query, the perturbation generator generates entirely distinct random perturbations. This is because attack queries are nearer to decision boundary,  substantiated by [2]. As a result, the prediction class probabilities are perturbed more significantly with input perturbation.
>
> **Q 1.3**: defense assumes attack query OOD?
>
> **A 1.3**: Thanks for your questions.
> As in earlier clarifications,  DFME attack query data comprises synthetic out-of-distribution (OOD) data. This is consistently substantiated by a range of existing studies [1,2,3,4]. Due to the attacker's ability to leverage diverse and appropriate data distributions for querying the target model, the exact distribution of these query data remains unknown to defender. Our proposed method perturbs these data and effectively ensures the preservation of model utility and attaining an excellent defense performance. The principles are  illustrated in **Q 1.1** and **A 1.1**.
>
> Furthermore, our method **does not rely on the assumption that attack query is OOD** and can be viewed as random smoothing: $T_{\theta_T}^{S}(x) = \int T_{\theta_T}(x + h_{\omega}(x, \epsilon)) d\epsilon$, where $\epsilon \sim N(0, I)$. Compared to learning a single function $T_{\theta_T}(x)$, learning the smoothed function $T_{\theta_T}^{S}(x)$ necessitates a larger number of queries for attacker due to the requirement of calculating the integral. By necessitating a larger query budget, our approach makes it considerably harder for attackers to successfully clone the model.
>
> **Q 2**: I think OOD attack queries will surely degrade the attack performance.
>
> **A**:  We would like to clarify that successful model extraction **does not necessarily require in-distribution query data**. As evidenced by existing DFME  attacks [3, 4], they employ out-of-distribution (OOD) data for querying the target model. Despite the utilization of such OOD data, these attacks still closely replicated the functionality of target model without degrading attack performance.
>
> **Q3**: theoretical guarantee
>
> **A**:  We invite you to refer to **global response** for theoretical analysis. We provide **Figure 1 in rebuttal.pdf in the global response** for visualization.
>
> Reference:
>
> [1] Increasing the Cost of Model Extraction with Calibrated Proof of Work. ICLR 2022
>
> [2] Zero-Shot Knowledge Distillation from a Decision-Based Black-Box Model. ICML 2021
>
> [3] Data-Free Model Extraction. CVPR 2021
>
> [4] Towards Data-Free Model Stealing in a Hard Label Setting. CVPR 2022
>
> [5] Protecting DNNs from Theft using an Ensemble of Diverse Models. ICLR 2021

---

> > ### Comment · Reviewer_VEKG · 2023-08-17
> >
> > Thank you for the clarification and details. Please make sure to include the promised revisions in the paper.

---

> > > ### Author Response · Authors · 2023-08-17
> > > **Thanks for your response**
> > >
> > > Thank you for getting back to us. We will incorporate the revision as promised in the final version.

---

### Author Rebuttal · Authors · 2023-08-07

# Global Response (Theoretical Analysis)
Given the random perturbation applied to the query input, the loss function for extracting the victim model becomes noisy. Attacker has to employ the following noisy loss function to extract the target model.

$\mathcal{L}_{C}^{\Delta}(\theta_C) := KL(T(x + h^{\omega}(x, \epsilon); \theta_T), C(x;\theta_C))$

where $T(x, \theta_T)$ is the target victim model with parameters $\theta_T$, $C(x;\theta_C)$ is the clone model with parameters $\theta_C$ and  $h_{\omega}(x, \epsilon)$ is the data-dependent perturbation. $KL$ denotes the KL divergence

The ground truth loss function without input perturbation (which remains inaccessible to the attackers due to their lack of knowledge regarding the amount of input perturbation) is illustrated below.

$\mathcal{L}_{C}(\theta_C) :=   KL(T(x; \theta_T), C(x;\theta_C))$

Ideally, attacker should optimize the following loss.

$\mathcal{L_C}^{*} = \min_{\theta_C} \mathcal{L}_{C}(\theta_C)$

 Due to the noisy loss function, attacker loss gradient becomes noisy and biased, shown as the following:

 **Attacker Loss Gradient Modeling**:

$G_t(\theta_C) = \nabla_{\theta_C} \mathcal{L_C}^{\Delta}(\theta_C) = \nabla_{\theta_C} \mathcal{L}_{C}(\theta_C) + B_t(\theta_C) + N_t(\theta_C)$

where $\nabla_{\theta_C} \mathcal{L}_{C}^{\Delta}(\theta_C)$ is the actual gradient adopted by the attacker

$\nabla_{\theta_C} \mathcal{L}_{C}(\theta_C)$ is the ground truth loss gradient without input perturbation, which is inaccessible to attacker

$B_t(\theta_C)$ is the gradient bias due to the noisy loss function introduced by perturbation generator $h_w$

$N_t(\theta_C)$ is the random variable introduced by the randomness in data samples. For analysis convenience, the expectation of the noise is assumed to be zero, i.e. $\mathbb{E} N_t(\theta_C) = 0$

where attacker updates the clone model as $\theta_C^{t+1} = \theta_C^t - \alpha G_t$; $\alpha$ is learning rate

**Assumption 1**: We assume Polyak- Łojasiewicz (PL)
condition on attacker loss function $\mathcal{L}_{C}(\theta_C)$

$|\nabla_{\theta_C} \mathcal{L_C}(\theta_C)|^2 \geq 2\omega (\mathcal{L_C}(\theta_C))$

**Assumption 2**
We assume the following smoothness condition for the loss function  $\mathcal{L}_{C}(\theta_C)$ by following [2]:

$\mathcal{L}{C}(\theta_C^2) \leq \mathcal{L}{C}(\theta_C^1) + \langle \nabla_{\theta_C} \mathcal{L}_{C}(\theta_C^1), \theta_C^2 - \theta_C^1 \rangle + \frac{A}{2} |\theta_C^2 - \theta_C^1|^2$

**Assumption 3** Assume bounded gradient bias and randomness

$\mathbb{E}||N_t(\theta_C)||^2 \leq D ||\mathcal{L}_{C}(\theta_C) + B_t(\theta_C)||^2 + \rho^2$

$||B_t(\theta_C)||^2 \leq d||\nabla_{\theta_C} \mathcal{L}_{C}(\theta_C)||^2 + \tau^2$ ($0 \leq d<1$)

where $\omega, D, d, \rho, \tau$ are constants

**Theorem 1**. Assume the attacker loss $\mathcal{L}_{C}(\theta_C)$ function satisfy assumption 2 and 3. $\alpha \leq \frac{1}{(D+1)A}$. Then, the attacker loss function satisfies the following inequality:

$\mathbb{E}[\mathcal{L_C}(\theta_C^{t+1})] \leq \mathcal{L_C}(\theta_C^{t}) + \frac{\alpha}{2}(d-1)|\nabla_{\theta_C} \mathcal{L}_{C}(\theta_C)|^2 + \frac{\alpha}{2}\tau^2 + \frac{\alpha^2 A \rho^2}{2}$

**Proof**: By assumption 2 and 3. We set $\theta_C^{1} = \theta_C^{t}$ and $\theta_C^{2} = \theta_C^{t+1}$ with $\theta_C^{t+1} = \theta_C^t - \alpha G_t$

$\mathbb{E}[\mathcal{L_C}(\theta_C^{t+1})] \leq \mathcal{L_C}(\theta_C^{t}) -\alpha \langle \nabla_{\theta_C} \mathcal{L}_{C}(\theta_C), \mathbb{E}(G_t) \rangle + \frac{\alpha^2A}{2} \left(\mathbb{E}|G_t - \mathbb{E}G_t|^2 + \mathbb{E}|\mathbb{E}G_t|^2\right)$

$= \mathcal{L_C}(\theta_C^{t}) -\alpha \langle \nabla{\theta_C} \mathcal{L_C}(\theta_C), \mathbb{E}(G_t) \rangle + \frac{\alpha^2A}{2} \left(\mathbb{E}|N_t|^2 + \mathbb{E}| \nabla_{\theta_C} \mathcal{L_C}(\theta_C) + B_t|^2\right)$

$\leq \mathcal{L_C}(\theta_C^{t}) -\alpha \langle \nabla{\theta_C} \mathcal{L_C}(\theta_C), \mathbb{E}(G_t) \rangle + \frac{\alpha^2A}{2} \left((D+1)\mathbb{E}|  \nabla_{\theta_C} \mathcal{L_C}(\theta_C) + B_t|^2 + \rho^2\right)$

$\mathbb{E}[\mathcal{L_C}(\theta_C^{t+1})] \leq \mathcal{L_C}(\theta_C^{t}) + \frac{\alpha}{2} \left(-2 \langle \nabla_{\theta_C} \mathcal{L_C}(\theta_C), \mathbb{E}(G_t) \rangle + |\nabla{\theta_C}\mathcal{L}_{C}(\theta_C) + B_t|^2 \right) + \frac{A\alpha^2\rho^2}{2}$

$= \mathcal{L_C}(\theta_C^{t}) + \frac{\alpha}{2} (-|\nabla{\theta_C} \mathcal{L}_{C}(\theta_C)|^2 + |B_t|^2) + \frac{A\alpha^2\rho^2}{2}$

$\leq \mathcal{L_C}(\theta_C^{t}) + \frac{\alpha}{2}(d-1) |\nabla{\theta_C} \mathcal{L}_{C}(\theta_C)|^2 + \frac{A\alpha^2\rho^2}{2}$

**Theorem 2** With assumption 3, convergence error of attacker loss function can be estimated as the following:

$L_C^{T}  \leq (1 - \alpha \omega(1-d))^T L_C^{0} + \frac{\tau^2 + A\alpha \rho^2}{\omega(1-d)}$

**Proof**:  We define $L_C^t = \mathcal{L_C}(\theta_C^t) - \mathcal{L}_{C}^{*}$. Then, we apply assumption 1 to theorem 1 and got following:

$L_C^{t+1} = (1 - \alpha \omega(1-d)) L_C^{t} + \frac{\alpha}{2}\tau^2 + \frac{\alpha^2A\rho^2}{2}$

We set $\kappa = \frac{\tau^2 + A\alpha \rho^2}{\omega(1-d)}$.
 Above equation can be rearranged as following :

$L_C^{T} - \kappa \leq (1 - \alpha \omega(1-d))^T(L_C^{0} - \kappa)$

Therefore, $L_C^{T}  \leq (1 - \alpha \omega(1-d))^T L_C^{0} + \frac{\tau^2 + A\alpha \rho^2}{\omega(1-d)}$

**Remark** From the above analysis, it becomes evident that accumulation of gradient estimation errors leads to a deviation of the final estimation error $L_C^{T} := \mathcal{L_C}(\theta_C^T) - \mathcal{L_C}^{*}$ from the ground truth.
The first term in the above inequality $(1 - \alpha \omega(1-d))^T L_C^{0} \rightarrow 0$.
This deviation occurs due to the increase in $\frac{\tau^2 + A\alpha \rho^2}{\omega(1-d)}$ caused by higher gradient bias $\tau$ and gradient randomness $\rho$.  Consequently, extracted model by attacker has a larger optimization error.

---

### Comment · Area_Chair_YzGT · 2023-08-16
**Discussion**

Dear Reviewers,

I appreciate your efforts thus far. Please read the author's rebuttals and other reviews attentively and respond to at least acknowledge that you've seen them. If your evaluation of the paper changes, please update your score and briefly explain the difference.

The paper received diverging initial reviews. Please consider discussing whether we can reach a consensus with the authors or other reviewers.

Thank you,
AC

---

### Comment · Area_Chair_YzGT · 2023-08-19
**Look forward to further feedback**

Dear Reviewers,

The open discussion phase of the paper is nearing its end, and the authors have provided detailed explanations in the rebuttal phase focusing on the theoretical guarantees, performance tradeoffs, and experimental details of the proposed algorithms. In order to ensure the smooth running of the conference, we would like to receive your responses to the authors' rebuttals as early as possible. Therefore, we kindly request you to submit your feedback as early as possible, if possible. Once again, we thank you for your time and look forward to your valuable comments.

Thank you,
AC

---

### Decision · Program_Chairs · 2023-09-21

**Decision:**

Accept (poster)

**Comment:**

The paper points out three problems with data-free model extraction defense methods, namely inefficiency, strong assumptions, and untimely. In order to solve the above problems, the author proposes a defense method called MeCo, which can prevent the above model extraction attacks by achieving efficient distributed robust training while maintaining the practicality of the model. In a rebuttal, the author's response provides a detailed theoretical analysis as well as a hyperparameter analysis, convincing the reviewers to update their assessment. Due to providing more insights and theoretical explanations in the rebuttal, the AC suggested accepting this paper. The AC had a thorough discussion with the SAC and the SAC agreed on the decision. Detailed theoretical explanations about the paper and related experiments in different attack scenarios need to be further supplemented. Authors are encouraged to consider the reviewers' detailed comments and clearly present the guiding theory in the final version.